# Administration of Levetiracetam via Subcutaneous Infusion for Seizure Control in the Palliative Care Setting: A Narrative Review

**DOI:** 10.3390/pharmacy12040125

**Published:** 2024-08-16

**Authors:** Fern Beschi, Rachel Hughes, Jennifer Schneider

**Affiliations:** 1Discipline of Clinical Pharmacology, College of Health, Medicine and Wellbeing, University of Newcastle, Callaghan 2308, Australia; jennifer.schneider@newcastle.edu.au; 2Department of Palliative Care, Calvary Mater Hospital Newcastle, Waratah 2298, Australia

**Keywords:** levetiracetam, palliative care, seizure, subcutaneous

## Abstract

This narrative review aims to summarise the information available on the use of subcutaneous (SC) levetiracetam (LEV) in the adult palliative care setting using clinical texts, databases, journals, and grey literature. A search strategy utilising Embase, Medline CINALH and Cochrane databases, as well as Google Scholar, was conducted with the mapped search terms “levetiracetam”, “subcutaneous” and “palliative”. LEV intravenous (IV) proprietary products are used subcutaneously, including as continuous subcutaneous infusions (CSCIs), in the adult palliative care setting. The total LEV daily dose ranged from 250 mg to 5000 mg and LEV was administered with various diluents at varying volumes. The data identified a clinical desire to mix LEV with other medications; however, the current evidence on combination compatibility is observational only and drug stability in combinations is lacking. The majority of information in the literature on SC LEV use is based on case reports and retrospective audits. Case reports, whilst at times offering more clinical detail, represent specific circumstances not necessarily applicable to a larger patient cohort. The findings of retrospective audits are limited by the documentation and detail reported at the time of patient care that may not be designed for data collection.

## 1. Introduction

Palliative care is defined by the World Health Organisation as “an approach that improves the quality of life of patients and their families who are facing challenges associated with life threatening illness”; its integration is recognised as a human right in health care [1]. The palliative approach prioritises patient and carer-centred, quality of life care where physical, social, psychological, and spiritual needs are addressed, cognisant of the goals of the patient and family. The paradigm of palliative care has evolved since its inception from care during the dying phase of life to working collaboratively with primary care and other specialties in managing complex symptomatology including during curative treatment attempts for life-threatening illness [2].

Levetiracetam (LEV) is an antiepileptic drug (AED) indicated for the prevention and management of focal (partial), generalised (tonic–clonic) or myoclonic seizures [3,4]. Unlike other AED, LEV is not thought to bind to the GABA, NMDA or benzodiazepine receptor sites [5]. While the exact mechanism is yet to be elucidated, it is theorised that LEV interacts with Synaptic Vesical Protein 2A (SV2A) on synaptic neurons, impairing the release of neurotransmitters. It is also thought to reduce the release of intraneuronal calcium and partially inhibit N-type calcium channels [6].

Patients receiving palliative care for life-limiting illnesses may experience seizures in a number of contexts, including pre-existing epilepsy or primary and secondary brain tumours; such patients often require antiepileptic drugs to reduce the occurrence and impact of seizures [7,8]. LEV has several characteristics that have rendered it a favourable option in palliative care. It is primarily eliminated by renal excretion, although one-third of the drug is metabolised by non-hepatic hydrolysis [8]. Oral levetiracetam has been reported to have rapid and almost complete absorption with a bioavailability of >95% and elimination half-lives in adults ranging from 6 to 8 h and 10 to 11 h in elderly volunteers [9]. When levetiracetam was administered as a short-term IV infusion (1500 mg over 15 min) to Japanese and Caucasian volunteers, the plasma drug concentration–time profile was observed to follow a biphasic decline. The observed mean clearance (3.43 L/h and 3.58 L/h, respectively) and volume of distribution after IV infusion (35.691 L and 41.17 L, respectively) were similar in both groups [10]. A population pharmacokinetic study in adult patients with epilepsy receiving oral levetiracetam estimated the apparent clearance (CL/F) and volume of distribution (V/F) to be 3.9 L/h and 65.3 L, respectively. Body weight was reported to be a significant covariate for CL/F and V/F, while the glomerular filtration rate significantly affected CL/F [11]. A recent systematic review of the pharmacokinetics of levetiracetam also identified weight and renal function as covariates most likely to influence the pharmacokinetics of levetiracetam [12]. LEV has few drug–drug interactions, which is an important consideration in palliative care, where patients may be prescribed multiple medications in evolving pharmacokinetic profiles [13]. LEV can be used when other AEDs may interact in combination, such as Sodium Valproate, or are best avoided due to comorbidities, including Carbamazepine in severe hepatic impairment or cardiac conditions such as lacosamide.

In the palliative setting, the oral route of medication administration may prove non-feasible in a number of contexts including, but not limited to, uncontrolled nausea and vomiting, altered gastrointestinal absorption, swallowing difficulties due to direct effects of head and neck malignancies or metastatic disease or during reduced consciousness. In these contexts, alternative routes of drug administration are required to ensure ongoing symptom management and patient comfort. In palliative care, continuous subcutaneous infusions (CSCIs) using subcutaneous pump devices such as syringe drivers may be used for this purpose. These small portable devices may be used by ambulant patients, enabling mobility, and removing the need to be attached to an intravenous (IV) pole. For patients experiencing concurrent symptoms, including pain, nausea, vomiting, and breathlessness, it may be desirable to combine drugs into one continuous subcutaneous infusion.

LEV is available as both and oral and parenteral formulations. The product information for parenteral LEV recommends that it is administered diluted in at least 100 mL of a compatible diluent as a 15 min IV infusion [14]. In the palliative care setting, in the interest of symptom management and comfort, medications may be administered in a way that deviates from their marketing authorisation (often referred to as “off-label”) [15]. At this time, the practice of subcutaneous infusion of LEV falls into this category. When administered by CSCI, the LEV infusion solution may be at higher concentrations [16], may be admixed with diluents different to those recommended by pharmaceutical manufacturers, be infused over longer periods (12–48 h) or be exposed to a broader range of temperatures, i.e., in the setting of a syringe driver located in bed next to the patient. These factors may influence the stability of LEV and affect the actual dose delivered, with potential impact on therapeutic effect or toxicity.

Limited data are available about the compatibility and stability of LEV for infusion when combined with other drugs. LEV infusion solution contains acetate buffer that adjusts the pH to 5.5. It is common practice for products to be adjusted to an optimal pH for stability. Mixing other drug solutions into a buffered solution may result in a change in pH that may influence stability and compatibility. When admixtures of drugs are delivered in the same infusion, it is essential to ensure compatibility and prevent the precipitation of the drug or other excipients. Precipitants may reduce the dose given, block infusion lines and may produce chemical compounds with no or different pharmacological effects [17]. Precipitants may be visible to the naked eye or sub visual particles requiring analytical equipment to detect.

When drugs are used “off-label”, as is the case for CSCI LEV, the lack of published data to inform safe use and patient safety warrants consideration of the development of recommendations and guides for clinical use. This review examines the published literature on using CSCI of LEV in the palliative care setting. It provides valuable information to inform the development of clinical guidance for using CSCI LEV where currently little scientific evidence is available. It will also outline the current knowledge on the stability and compatibility of LEV and drug combinations and identify gaps in knowledge that need to be addressed through future research.

## 2. Literature Search Strategy

A literature review was undertaken using Embase, Medline, CINAHL and Cochrane databases with the subject terms and mapped phrases “Levetiracetam”, “Subcutaneous” and “Palliative”. A further search using Google Scholar was completed where grey literature was found and included. The results were limited to the adult palliative population and only papers written in English were included. There were no exclusions based on article type and letters to the editor and conference abstracts were included. Two authors J.S. and F.B read the titles and abstracts to screen for relevance. Selected articles were read, and applicable information was extracted. An additional search was performed to identify studies investigating the stability and compatibility of LEV alone and in combination with other medications in solutions. Embase, Medline and CINAHL databases as well as Google Scholar were utilised with the search terms “Levetiracetam” and “Stability” or “Compatibility”. This secondary search had no exclusion criteria; however, only the first five pages of the Google Scholar search results were reviewed.

## 3. Subcutaneous Levetiracetam Overview

### 3.1. Efficacy, Dosing, Administration Regimens and Side Effects

In a systematic review, Koekkoek et al. reported that the use of a non-enzyme inducing antiepileptic drug after the first seizure in patients with brain tumours is strongly recommended [18] and that LEV is the most frequently prescribed first-line antiepileptic drug [19]. Further, when the efficacy of a range of different antiepileptic drugs in patients with glioma and epilepsy was compared, LEV had high efficacy as monotherapy and the lowest treatment rate failure [20]. About half of the adverse effects observed with LEV in patients with glioma were of psychiatric origin and therefore not recommended in patients with a history of psychiatric disorders such as anxiety and depression. LEV is reported to have side effects including drowsiness, fatigue, risk of agitation and emotional lability [21].

The European Association of Neuro-Oncology recently published guidelines for managing neurological and vascular complications of brain metastases and primary brain tumours. It was noted that there are no randomised trials in this population and after considering the risks and benefits of classic and newer antiepileptic drugs, LEV and lamotrigine were recommended as preferred medications [22].

Table 1 shows the reported SC use of LEV in the adult palliative care population. Single-patient case reports have discussed the successful use of LEV administered subcutaneously [23,24,25]. Mas-Sese et al. described the experience of using subcutaneous (SC) LEV in six patients receiving palliative care when changed to SC administration when oral LEV administration was no longer possible [26]. Administration was intermittent subcutaneous infusion (every 12 h), and the drug was diluted in 100 mL saline solution. Doses administered every 12 h ranged from 500 to 1500 mg. The mean duration of SC administration was 9.5 days. None of the patients had seizures after the onset of SC LEV, although the frequency of seizures for each patient prior to SC LEV varied from 7 in the last 15 days to 1 in 3 months.

Several retrospective audits/reviews of the use of CSCI LEV were identified. Cran et al. conducted a regional multi-centre audit of prescribing practice in palliative care in patients unable to take oral medications and requiring seizure management [27]. Of the 26 patients included in the audit, 16 were prescribed LEV CSCI in doses ranging from 250 mg to 3 g over 24 h, highlighting that LEV was commonly used in this clinical setting.

Twigger et al. conducted a retrospective audit where 31 cases of LEV use were identified [28]. In these cases, 42% of patients had seizures in the week prior to commencing with 58% described as tonic–clonic seizures, while nearly 93% had antiepileptic drugs prescribed prior to commencing. Most patients (92%) received LEV as CSCI, with a few receiving a bolus SC dose. Doses at commencement ranged from 250 to 3000 mg, and 12% were titrated over time due to seizure activity. Overall, 81% reported no side effects, and 16% reported local skin reactions. No further seizures were documented for 70% of patients on LEV SC, and 62% of patients remained on SC LEV until death. Remi et al. reported good efficacy, tolerance and an 80% success rate in controlling epileptic events using CSCI LEV, although 15 of the 20 patients identified through the retrospective chart review were co-administered midazolam [16]. The LEV total daily doses ranged from 960 mg to 4 g. Of the patients identified, 60% were on oral LEV prior to starting CSCI LEV, and 25% were not on any AED prior to the CSCI LEV. Kondasinghe et al. conducted a retrospective data analysis of SC infusions of LEV [29]. In the eight patients identified, LEV dosing ranged from 500 to 3000 mg/day, and LEV infusion concentrations ranged from 20 to 83 mg/mL. The median duration of administration was 6.5 days and there was no seizure recurrence in 75% of patients. No adverse effects were reported. Gaviria-Carrillo et al. described their clinical experience with SC LEV and conducted a retrospective review [30]. Doses administered ranged from 1000 to 3000 mg daily, and the principal diagnosis was structural focal epilepsy. The duration of use of SC LEV ranged from 1 day to 360 days. The authors observed good tolerability and effectiveness in controlling seizures.

Different volumes of diluent used to administer CSCI LEV are reported in the literature. Some publications report using 100 mL volumes [31,32], while others use smaller volumes (40–50 mL) [21,30]. Volume is dictated by the devices available for clinical use and will determine the concentration of the infusion solution being administered. The use of different diluents for LEV infusions is also reported. Commonly used diluents are 0.9% sodium chloride solution [23,26,32] and water for injection [33], with 5% glucose in water reported on a one occasion [25].

### 3.2. Tolerability at Subcutaneous Infusion Sites

LEV injection has high osmolarity (3610 mOsm/L), and manufacturers recommend diluting it in at least 100 mL of fluid for IV injection. When administered via CSCI, higher concentrations may occur due to the limited volume of devices. Wells et al. reported no SC site discomfort when LEV was administered as 2 g in 100 mL of 0.9% sodium chloride solution over 24 h [24]. Remi et al. reported good local SC site tolerability in 18 out of 20 patients receiving a high osmolarity solution (571–3597 mOsm/L) by CSCI [16]. Two patients experienced erythema at the injection site. In an audit and review, three out of seven patients who had site reactions were receiving a CSCI in which other drugs were mixed with LEV. In 19 patients receiving CSCI LEV diluted with WFI, none were reported to have local site reactions [21].

### 3.3. Oral to Subcutaneous Conversion

When commencing LEV, it is recommended to start at a low dose and slowly increase the dose. Since a major route of elimination is renal excretion, renal function should also be considered [14]. When changing routes of administration, the doses of drugs may need to be adjusted due to variability in absorption (bioavailability) and first-pass metabolism. The recommended dose conversion when switching LEV from oral to IV delivery is 1:1 [35]. No formal studies have been conducted to determine the conversion ratio from oral to subcutaneous infusion.

Monteiro et al. conducted a retrospective observational analysis [31]. This review identified four patients who were administered SC LEV using an SC: oral ratio of 1:1. These patients had the dose diluted in 100 mL of 0.9% sodium chloride and administered as a bolus with the dose ranging from 2000 to 5000 mg. Murray-Brown et al. published a case report of LEV CSCI using water for injection as the diluent [33]. While an initial 1:1 ratio was used to convert from an oral LEV dose to the SC LEV dose, the syringe contained multiple medications, including oxycodone and metoclopramide, and required an increase in LEV dose after 24 h due to upper limb twitching. This may indicate that LEV was not stable with this admixture or the use of a 1:1 conversion ratio resulted in sub-therapeutic dosing in the patient, or a combination of these factors. Neither oxycodone nor metoclopramide are known to reduce seizure thresholds.

Furtado et al. reported a patient who received IV LEV after the failure of phenytoin 300 mg/day [25]. The LEV dose was titrated until symptomatic control was achieved at 1 g/day (two divided doses, every 12 h). For ongoing management in the community, other AEDs options were trialled unsuccessfully. The patient was then commenced on SC LEV using an IV: SC conversion ratio of 1:1. The SC infusion consisted of LEV 1 g diluted to 750 mL of 5% glucose and administered at 31.25 mL/h over 24 h. The patient continued the SC infusion for four weeks and seizures were reduced to less than one per week.

The retrospective audit by Remi et al. identified eleven patients that switched from oral to SC LEV [16]. The doses administered by SC infusion were approximately the same as the daily oral doses administered in five patients. One patient had IV LEV; for the remaining six patients, the CSCI LEV doses were mostly higher than oral doses. No plasma drug concentration data are available and the factors contributing to requiring higher doses are not known.

### 3.4. Pharmacokinetics of LEV Administered by SC Infusion

Since CSCI LEV is currently “off-label”, the required pharmacokinetic studies for market authorisation to determine drug absorption and clearance are not available. Westphal et al. reported a case study where a patient was converted from oral to CSCI. At a dose of 1500 mg/24 h, seizures were controlled in this patient and the serum LEV concentrations achieved were within or close to the reference therapeutic range (118–235 umol/L) [34]. In a recent study, Papa et al. investigated plasma concentrations of LEV in seven patients receiving SC LEV administered as a 30 min infusion every 12 h [32]. The reported mean plasma LEV concentration was 14.4 mg/L at 1000 mg/day and 27.7 mg/L at 2000 mg/day. The results suggest linear pharmacokinetics with a proportional increase in the steady-state concentration with the dose. The authors performed population pharmacokinetic analysis using these data and reported a clearance (CL/F) of 2.5 L/h and a terminal half-life of 10.4 h for LEV administered by the SC route as a short infusion every 12 h. The reported CL/F and terminal half-life following SC infusion are in a similar range to that reported after short-term IV infusion [10] and oral administration [9]. While this study does show systemic absorption of LEV and linear pharmacokinetics with SC administration, the plasma LEV concentration profile over time will differ from that for a CSCI.

### 3.5. Stability of LEV in Infusion Solutions

Limited data are available in the literature on the stability of LEV when diluted and prepared as an infusion solution. A study by Raphael et al. investigated the stability of LEV prepared at a concentration of 1000 mg/25 mL in 0.9% sodium chloride solution and stored in PVC bags, polyolefin bags and polypropylene syringes at 2–8 °C [36]. The solution was observed to be physically and chemically stable for up to 14 days. Solutions were not tested at room temperature or higher temperatures that may be experienced in clinical use.

### 3.6. Combinations and Compatibility

Commonly used resources such as Trissel’s^TM^ [37], which pharmacists use to check stability and compatibility, mainly focus on IV delivery of parenteral medication and admixtures, usually over a short duration, e.g., up to 4 h. This is not relevant in the palliative care setting.

The literature search resulted in only three publications that conducted physical compatibility studies with parenteral LEV. These studies investigated the compatibility of LEV with heparin, dobutamine and dopamine [38] using an all-in-one parenteral nutrition solution [39] and a further study investigated Y site compatibility with piperacillin-tazobactam, cisatracurium, dexmedetomidine, fosphenytoin, norepinephrine, vasopressin, vancomycin and propofol [40]. While these studies do contribute to the overall knowledge on compatibility, these applications do not align with combinations likely to be prescribed in palliative care contexts and the extent of time for which drugs are mixed.

Case reports and retrospective audits on LEV use in palliative care have mentioned LEV being combined with metamizole, midazolam, morphine, hyoscine butylbromide, hydromorphone, methotrimeprazine, metoclopramide, dexamethasone, haloperidol, glycopyrrolate and clonidine [16] and morphine, midazolam, metoclopramide and dexamethasone [28]. Other reference sources acknowledge that compatibility data are lacking but clinical experience suggests it may be compatible with glycopyrronium, haloperidol, hyoscine butylbromide, levomepromazine, methadone, metoclopramide, midazolam, morphine sulfate, oxycodone and dexamethasone at a dose of 0.5 mg [35].

The extent and heterogeneity of these data illustrate the clinical desire to mix drugs with LEV for CSCI. However, the lack of robust scientific, therapeutic and safety data to underpin this practice is also highlighted. Further, the information available in the literature does not specify the concentrations of each drug mixed, the duration of infusions or what diluents were used in the admixture. The temperatures to which admixtures are exposed will vary according to practice but also the geographical location and time of year are not known.

Reported use of CSCI LEV is mostly based on retrospective audits or case reports. Ideally, future prospective studies and randomised controlled trials would be the best form of evidence. However, in the palliative care setting, randomised controlled trials are often challenging to conduct [41].

## 4. Discussion

The reporting of LEV in the literature has evolved over time from case reports to retrospective studies to prospective investigation into the pharmacokinetics of SC LEV. While the literature gives an overall picture of how LEV is being administered in a clinical setting, there is inadequate detail in the reporting to apply the information more broadly from the retrospective research found. For example, Monteiro et al.’s letter to the editor [27] refers to administering 100 mL bolus doses SC; however, the detail of administration duration/time for these doses could not be confirmed. Much of the data reported LEV used concurrently with other antiepileptic agents, meaning the efficacy of the drug cannot be established, or LEV was combined with other medications in a single syringe where the stability and compatibility were not guaranteed. Only a single prospective study on the pharmacokinetic profile of SC LEV [32], and one stability study of LEV infusion solution [36] were found. The former established the pharmacokinetics profile of SC LEV 30 min infusions, yet this profile will differ from that of LEV as a CSCI commonly used in practice. Likewise, the stability of LEV at a single concentration and low temperature will not be indicative of the stability and higher temperatures or a wider range of concentrations administered in a clinical setting.

Some publications provide data on drugs being mixed with LEV and co-administered with LEV without reference to the compatibility/incompatibility observations of the mixture (e.g., [16,33]), while others lacked information on the dose of other drug being combined with LEV [42]. Concentration of both drugs when mixed is essential as the concentration present may affect the stability and compatibility to different extents.

No studies investigating the stability of infusion solutions in different diluents, temperatures such as room temperature, 25 °C and at 37 °C (when infusion may be carried out in bed next to patient) and over time intervals of 24 h have been conducted. This highlights the need for well-designed stability studies to be conducted to inform current clinical practice. Similarly with compatibility, while observations of a precipitate or cloudiness will indicate incompatibility, sub-visual particles may also occur. Mixing drugs may also produce changes in pH, which can result in faster degradation of one or more drugs present.

Best practice would require the compatibility and stability of these admixtures to be confirmed but limited, if any, data are available to support this. Stability and compatibility studies require access to specialised analytical equipment and expertise and often there is a lack of interest in funding this research. Pharmacists are in a unique position to advocate for the importance of this type of research. Further research to investigate the pharmacokinetics of CSCI levetiracetam and the effect of factors including weight and renal function would assist in determining the best dosage regimens in this population.

## 5. Conclusions

With the availability of a parenteral formulation and its favourable profile for patients receiving palliative care, LEV is evidently being utilised as a CSCI over 24 h for seizure management in the palliative care setting. The reported doses administered range from 250 mg to 4000 mg in 24 h delivered in differing volumes and diluents, although not all reports directly addressed the volume delivered or diluent used. Anecdotally, good efficacy of LEV via this route is generally reported. While some site reactions are occasionally reported, in general, the drug appears to be well tolerated. However, it should be noted that LEV injection has high osmolarity and the lack of detail in volumes and diluents used makes it difficult to determine if the site reactions that do occur may be related to the high osmolarity of infusion solutions.

Best evidence for safe and effective use of medicine is mandated in all health care settings and equitable care principles encourage us to ensure the same for people receiving palliative care. Clinicians need to confidently address the complex symptomatology for patients receiving palliative care in the knowledge that they are acting safely and with best evidence. Doctors, nurses, and pharmacists engaged in the care of patients with palliative needs are currently disadvantaged by suboptimal data for clinical decision making for LEV SC infusions alone and with other drugs. Drug admixture studies are an under-conducted but critical area of research in palliative care. Further research using well-designed compatibility studies employing robust scientific techniques are required in clinical palliative care practice. These would include research to determine and measure concentrations of drugs using stability-indicating assays.

## Figures and Tables

**Table 1 pharmacy-12-00125-t001:** Summary of subcutaneous levetiracetam reported in the literature.

Paper	Number of Patients	Total Daily SC Dose Range	Method of SC Administration
Remi et al. (2014) [16]	20	960–4000 mg	CSCI
López-Saca et al. (2013) [23]	1	2000 mg	100 mL infusions twice a day
Wells et al. (2016) [24]	1	2000 mg	CSCI
Furtado et al. (2018) [25]	1	1000 mg	CSCI
Mas-Sese et al. (2021) [26]	6	1000–3000 mg	100 mL infusions twice a day
Cran et al. (2018) [27]	16	250–3000 mg	CSCI
Twigger et al. (2018) [28]	31	250–3000 mg	CSCI or intermittent infusions
Kondasinghe et al. (2022) [29]	8	500–2000 mg	CSCI
Gaviria-Carrillo et al. (2023) [30]	21	1000–3000 mg	50 mL infusions two- or three-times day
Monteiro et al. (2022) [31]	4	2000–5000 mg	100 mL infusions two- or three-times day
Papa et al. (2021) [32]	7	1000–3000 mg	100 mL infusions twice a day
Murray-Brown et al. (2016) [33]	1	1500–2000 mg	CSCI
Westphal et al. (2021) [34]	1	1500 mg	CSCI

## Data Availability

No new data were created or analyzed in this study. Data sharing is not applicable to this article.

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
