# Peer review of "Administration of Levetiracetam via Subcutaneous Infusion for Seizure Control in the Palliative Care Setting: A Narrative Review"

_pharmacy, 2024, doi:10.3390/pharmacy12040125_

Round 1

Reviewer 1 Report

Comments and Suggestions for Authors

The article has been corrected as recommended.

Author Response

Comment 1: Please ensure you have defined abbreviations within the main text (e.g., SC, CSCI, LEV).

Response 1: Thank you for pointing this out. I have made these changes as highlighted in the text on lines throughout the text

Comment 2: Please ensure consistent use of abbreviations (LEV and levetiracetam is used interchangeably throughout the text).
Response 2: Thank you for pointing this out. I have made these changes as highlighted in the text.

Comment 3: Please rephrase sections of this paper to avoid reducing individuals to their medical conditions (e.g., rephrase "glioma patients" to "patients with gliomas").

Response 3: Thank you for pointing this out. I have made these changes as highlighted in the text.

Reviewer 2 Report

Comments and Suggestions for Authors

- The author aimed to summarise the information available on the use of subcutaneous (SC) levetiracetam (LEV) in the adult palliative care setting using clinical texts, databases, journals, and grey literature

- The article is timely and interesting and addresses a gap in the literature.

- The introduction is very well written.

- Methods are very well described

- Results are relevant and well presented.

- A Discussion section, before conclusions, should be added with a more critical insight. This section should link all the previous section of the article.

Author Response

Comment 1: A Discussion section, before conclusions, should be added with a more critical insight. This section should link all the previous section of the article.

 Response 1: Agree. We have, accordingly, added a discussion with a more critical insights and focus.

Reviewer 3 Report

Comments and Suggestions for Authors

Administration of Levetiracetam via subcutaneous infusion for seizure in the palliative care setting: a narrative review.

Off-label administration of a various agents is common nowadays. Thus, this topic can be interesting to discuss.

Introduction:

Lines 46-47: “It is primarily eliminated by renal excretion, although one-third of the drug is metabolised by non-hepatic hydrolysis”

I think the pharmacokinetic parameters should be discussed in more details. And I would put it in line 51 before discussing oral and parenteral routes of administration. It would be more logical.

I would describe all PK parameters, not only the route of elimination and metabolization, since the half-life, clearance and volume of distribution could be as important as elimination/excretion in CSCI administration! I can see that it is described in 3.4., but these are only drawn conclusions of studies. But we know exactly these parameters for oral and IV administration, and in this way reading the article we could make a comparison with SC/CSCI.

Lines 69-70: I would describe exactly how is performed a CSCI exactly, since the reader may not know..moreover, the absorbtion, etc is different from IV administration. (as you described in 3.3)

Materials and Methods:

Lines 106-108: “An additional search was performed to identify studies investigating the stability 106 and compatibility of levetiracetam alone and in combination with other medications in 107 infusion solutions.” I missed the exactly description: database-is similar as for previous search? What were the “mapped phrases” or keywords?

Results

Line 154: “concentrations ranged from 20 to 83mg/mL” I suggest to specify the type of concentration: infusion? Serum? making it clear for the reader.

Line 193: Was 100 ml LEV administered SC in a single bolus dose? It is very interesting. Please verify it..

Considering all together, after clarifying and making the corrections I would accept this manuscript for publication, since the article draws attention to the need f or additional research to stability/compatibility of LEV and defining PK parameters for CSCI.

Administration of Levetiracetam via subcutaneous infusion for seizure in the palliative care setting: a narrative review.

Off-label administration of a various agents is common nowadays. Thus, this topic can be interesting to discuss.

Introduction:

Lines 46-47: “It is primarily eliminated by renal excretion, although one-third of the drug is metabolised by non-hepatic hydrolysis”

I think the pharmacokinetic parameters should be discussed in more details. And I would put it in line 51 before discussing oral and parenteral routes of administration. It would be more logical.

I would describe all PK parameters, not only the route of elimination and metabolization, since the half-life, clearance and volume of distribution could be as important as elimination/excretion in CSCI administration! I can see that it is described in 3.4., but these are only drawn conclusions of studies. But we know exactly these parameters for oral and IV administration, and in this way reading the article we could make a comparison with SC/CSCI.

Lines 69-70: I would describe exactly how is performed a CSCI exactly, since the reader may not know..moreover, the absorbtion, etc is different from IV administration. (as you described in 3.3)

Materials and Methods:

Lines 106-108: “An additional search was performed to identify studies investigating the stability 106 and compatibility of levetiracetam alone and in combination with other medications in 107 infusion solutions.” I missed the exactly description: database-is similar as for previous search? What were the “mapped phrases” or keywords?

Results

Line 154: “concentrations ranged from 20 to 83mg/mL” I suggest to specify the type of concentration: infusion? Serum? making it clear for the reader.

Line 193: Was 100 ml LEV administered SC in a single bolus dose? It is very interesting. Please verify it..

Considering all together, after clarifying and making the corrections I would accept this manuscript for publication, since the article draws attention to the need for additional research to stability/compatibility of LEV and defining PK parameters for CSCI.

Author Response

Comment 1: I think the pharmacokinetic parameters should be discussed in more details. And I would put it in line 51 before discussing oral and parenteral routes of administration. It would be more logical.

I would describe all PK parameters, not only the route of elimination and metabolization, since the half-life, clearance and volume of distribution could be as important as elimination/excretion in CSCI administration! I can see that it is described in 3.4., but these are only drawn conclusions of studies. But we know exactly these parameters for oral and IV administration, and in this way reading the article we could make a comparison with SC/CSCI.

Lines 69-70: I would describe exactly how is performed a CSCI exactly, since the reader may not know..moreover, the absorbtion, etc is different from IV administration. (as you described in 3.3)

Response1: Thank you for pointing this out. We have included more information on pharmacokinetics of LEV via different administration routes and modes in adult populations.

Comment 2: Lines 106-108: “An additional search was performed to identify studies investigating the stability 106 and compatibility of levetiracetam alone and in combination with other medications in 107 infusion solutions.” I missed the exactly description: database-is similar as for previous search? What were the “mapped phrases” or keywords?

Response 2: Thank you for this comment on the oversight, more detail has been added to describe the secondary search added text highlighted  from line 125

Results

Comment 3 Line 154: “concentrations ranged from 20 to 83mg/mL” I suggest to specify the type of concentration: infusion? Serum? making it clear for the reader.

Response 3: Thank you for pointing this out. This has been changed to read LEV infusion concentrations.

Comment 4: Line 193: Was 100 ml LEV administered SC in a single bolus dose? It is very interesting. Please verify it.

Response 4: Thank you for pointing this out. I have confirmed this is all the detail provided in the original article however we have included this as part of the discussion see highlighted text line

Reviewer 4 Report

Comments and Suggestions for Authors

Nice and very useful paper .

Some hints :

Page 2, line 50 .

Which antiepileptics are contraindicated by  cardiac comorbidities , just phenytoin ? Please add a “such as “ both for liver and cardiac comorbidities citing the main drugs, so that it can be more useful for the reader.

Page 3 line 100

Please specify which kind of papers were included or excluded, case reports, RCT, reviews, book chapters .

Furthermore I wonder if there was any report concerning adolescents or children, or if this population was excluded from the search.

It would be of interest for the reader to have also a synthetic overview, how many total patients are reported to be treated according to the case series and reports of the review ? Which is the average success rate and the average adverse events rate ? 

Author Response

Comment 1:  Page 2, line 50 . Which antiepileptics are contraindicated by  cardiac comorbidities , just phenytoin ? Please add a “such as “ both for liver and cardiac comorbidities citing the main drugs, so that it can be more useful for the reader.

Response 1: Agree. We have, accordingly, included examples see highlighted text.

Comment 2: Page 3 line 100. Please specify which kind of papers were included or excluded, case reports, RCT, reviews, book chapters . Furthermore I wonder if there was any report concerning adolescents or children, or if this population was excluded from the search.

Response 2: Thank you for pointing this out. We have included more information regarding search parameters. See highlighted lines.

Comment 3: It would be of interest for the reader to have also a synthetic overview, how many total patients are reported to be treated according to the case series and reports of the review ? Which is the average success rate and the average adverse events rate ? 

Response 3: Thank you for pointing this out. We have included a table to summarise information and included in the discussion that we cannot infer the success of LEV alone as many times it was used in combination with other AEDs such as midazolam.